

# Habitat loss estimation for assessing terrestrial mammalian species extinction risk: an open data framework

Mariella Butti[1], Luciana Pacca[2], Paloma Santos[2,3,4], André C. Alonso[2], Gerson Buss[2], Gabriela Ludwig[2], Leandro Jerusalinsky[2] and Amely B. Martins[2]

[1] Centro Nacional de Pesquisa e Conservação de Mamíferos Carnívoros/CENAP, Instituto Chico Mendes de Conservação da Biodiversidade/ICMBio, Atibaia, São Paulo, Brazil
[2] Centro Nacional de Pesquisa e Conservação de Primatas Brasileiros/CPB, Instituto Chico Mendes de Conservação da Biodiversidade/ICMBio, Cabedelo, Paraiba, Brazil
[3] Instituto Nacional da Mata Atlântica/INMA, Santa Teresa, Espírito Santo, Brazil
[4] Instituto de Pesquisa e Conservação de Tamanduás no Brasil, Ilhéus, Bahia, Brazil

## ABSTRACT

Terrestrial mammals face a severe crisis of habitat loss worldwide. Therefore, assessing information on habitat loss throughout different time periods is crucial for assessing species' conservation statuses based on the IUCN Red List system. To support the national extinction risk assessment in Brazil (2016–2022), we developed a script that uses the MapBiomas Project 6.0 data source of land cover and land use (annual maps at 30 m scale) within the Google Earth Engine (GEE) platform to calculate habitat loss. We defined suitable habitats from the MapBiomas Project land cover classification for 190 mammalian taxa, according to each species range map and ecological characteristics. We considered a period of three generation lengths to assess habitat loss in accordance with the Red List assessment criteria. We used the script to estimate changes in available habitat throughout the analyzed period within the species' known ranges. The results indicated that habitat loss occurred within 94.3% of the analyzed taxa range, with the Carnivora order suffering the greatest habitat loss, followed by the Cingulata order. These analyses may be decisive for applying criteria, defining categories during the assessment of at least 17 species (9%), enriching discussions, and raising new questions for several other species. We considered the outcome of estimating habitat loss for various taxa when applying criterion A, which refers to population reduction, thus supporting more accurate inferences about past population declines.

## INTRODUCTION

Habitat loss and fragmentation significantly impact biodiversity, leading to critical population declines and affecting long-term biodiversity conservation (*Fahrig, 2003*; *de Barros Ferraz et al., 2021*; *Santos et al., 2019a*). Approximately 100 million hectares of

Corresponding author
Mariella Butti,
mariella.butti@icmbio.gov.br

rainforest were lost between 1980 and 2000 (*IPBES, 2019*), and the remaining are strongly affected by anthropogenic activities (*Haddad et al., 2015*; *Leblois, Damette & Wolfersberger, 2017*). Decreasing the amount of available habitat may directly affect critical biological processes, such as resource availability (*Ryser et al., 2019*), dispersal (*Cote et al., 2017*), pollination (*Pavageau et al., 2017*) and gene flow (*Dixo et al., 2009*; *Moraes et al., 2018*). Additionally, these activities contribute to the increased accessibility of hunters to natural/remote areas, resulting in higher direct exploitation of species (*Gallego-Zamorano et al., 2020*), and increasing the risk of invasive species introduction. Thus, habitat loss contributes to population reduction (*Harfoot et al., 2021*; *Heinrichs, Bender & Schumaker, 2016*; *McCormack, Ghani & Ferguson, 2019*).

Globally, 28% of all assessed species are threatened with extinction, corresponding to more than 40,000 species, of which 26% correspond to mammalian species (*IUCN, 2020*). Notably, in addition to being threatened by habitat loss, disturbance, and fragmentation (*IPBES, 2019*; *Bogoni, Peres & Ferraz, 2020*; *Canale et al., 2012*), these species have suffered from anthropogenic impacts and have been victims of several human-wildlife conflicts (*Adhikari et al., 2022*; *Desbiez, Oliveira & Catapani, 2020*; *Vanak & Gompper, 2010*). As mammalian species play a critical role in ecosystem functioning (*Jorge et al., 2013*; *Magioli et al., 2021*; *Rodrigues et al., 2019*), the continuous population reduction of diverse mammalian species over recent decades has directly impacted ecosystem dynamics (*IUCN, 2020*).

Reducing species extinction risk—particularly for highly threatened taxa—is a global priority and has been featured in various international agreements for biodiversity conservation, such as in the Convention on Biological Diversity Aichi Target 12 (either in its original form: https://www.cbd.int/sp/targets/, as well as in its post-2020 draft framework: https://www.cbd.int/doc/c/abb5/591f/2e46096d3f0330b08ce87a45/wg2020-03-03-en.pdf) and the United Nations Sustainable Development Goal 15 (https://sdgs.un.org/goals). In order to assess the extinction risk for known species, the International Union for the Conservation of Nature (IUCN) conducts global conservation status assessments using a well-established methodology and rigorous theoretical and analytical data (*IUCN, 2022*). Based on specific parameters and definitions (*e.g.*, population sizes and trends, geographic range and occupancy, population reduction, and generation time), the IUCN's assessment process has established the globally used extinction risk categories (*e.g.*, Critically Endangered, Least Concern, and Data Deficient), criteria (Criteria A to E) and assessment methodology (*IUCN, 2022*, *2012*).

Although there is a lack of primary biological data for many taxa (*IUCN, 2022*), the IUCN's methodology considers a variety of data types from many sources of varying quality. For example, the IUCN's Criterion A—widely used to assess mammalian species' risk of extinction—highlights taxa that have experienced intense population reductions either in the recent past or projected for the near future. However, direct observations of population reduction are not available for all taxa. Thus, other data types—such abundance indexes, declines in habitat quality, levels of exploitation, or effects of pathogens—may be used for estimation, inference, or suspicion of population reduction. Habitat loss estimates within a taxon's range, area of occupancy, or extent of occurrence

may be used as an essential tool for extinction risk assessments and as a proxy of population reduction, especially for species that are strictly related to or dependent upon their suitable natural habitat (*IUCN, 2022*).

Brazil is the fifth largest country in the world, with the greatest global biodiversity and is responsible for roughly 14% of the world's biota, even though many species are yet to be discovered and cataloged (*Costa et al., 2005*). There are around 770 mammalian species (*Abreu et al., 2021*) in Brazil, accounting for the second-greatest mammalian diversity in the world and the largest for some mammalian orders, such as Primates (*Mittermeier et al., 2013*) and Xenarthra (*Santos et al., 2019b*). Despite housing this great level of mammalian biodiversity, exceptional levels of deforestation affect mammal species in all Brazilian phytogeographic domains (*Bogoni, Peres & Ferraz, 2020*; *Magioli et al., 2021*), including the Amazon, which presents the greatest species richness; the Atlantic Forest, which presents increased rates of endemism (*Mittermeier et al., 2011*); and the Pantanal, which has lost approximately 17 million vertebrates over the past 2 years due to wildfires (*Tomas et al., 2021*). An area of approximately 82 Mha of natural habitats was lost in Brazil between 1985 and 2020, including a notable increase in deforestation during recent years in almost all phytogeographic domains, directly affecting multiple taxa (*Souza et al., 2020*; *MapBiomas Project, 2021*). The future trend is uncertain, with or without the compliance of Brazilian public policies such as the Brazilian Forest Code (*Brasil, 2012*). In a modeling study, *Soterroni et al. (2018)* presented a spatially explicit assessment of Brazil's 2012 Forest Code through the year 2050 with a detailed representation of the agricultural sector and spatial land-use change. Their results showed that, if rigorously enforced, the Forest Code could prevent a net loss of 53.4 Mha of forest and native vegetation by 2050, 43.1 Mha (81%) of which are in the Amazon alone.

Throughout the years, the high number of threatened species and unassessed taxa indicate the necessity for more efficient extinction risk assessments, including the production of new data and metanalyses to support IUCN categories and criteria application. Since land-use changes very fast globally, automatic methods with satellite images make the assessment of habitat loss more dynamic and efficient. Moreover, these methods may be a promising tool for supporting extinction risk assessments and in guiding decision-making (*de Barros Ferraz et al., 2021*). Furthermore, performing metanalyses with open data would ultimately contribute to greater reproducibility, replicability, and data cycling—the basis of the open science philosophy (*Gallagher et al., 2020*)—within the assessment process. South America currently depends on the MapBiomas Project, a collaborative network formed by NGOs, universities and technology startups (*MapBiomas Project, 2021*). The MapBiomas project has produced land use and land cover maps (LULC) since 1985—in addition to other products such as fire and deforestation reports. Within this project, LULC maps are updated yearly and include all Brazilian and many South American phytogeographic domains (Amazon, Chaco, Pampa and Atlantic Forest), in addition to recently mapping Indonesia. Along with the Google Earth Engine technology, a cloud-based platform with a vast satellite image collection catalog (*Gorelick et al., 2017*), the MapBiomas LULC maps may be used to produce land cover change estimates at a faster and more efficient rate, without requiring local

supercomputers. Therefore, we used a series of land-cover maps from the MapBiomas Project to generate a cloud-based, open data framework—using the Google Earth Engine platform (GEE)—that allows habitat loss estimation across different time scales.

This study established a systematic approach to estimating habitat loss for terrestrial species using a time series of land use classification maps to generate necessary, reliable, and prompt information in order to subsidize the application of the IUCN categories and criteria for extinction risk assessments. The novelty here is that we created a method to obtain an accessible and standardized analysis of habitat loss to assess species' risk of extinction, directly applicable to South American and Indonesian species but which may also be adapted for other tagged landcover Datasets on Earth Engine. The data generated is of great value to the process of qualifying species under the IUCN criterion A, which is based on population reduction estimates for the past, present, and/or projected for the future, considering the sub-criterion c (habitat loss), thereby strengthening the results, as well as informing conservation decision-making.

## MATERIALS AND METHODS

### Studied system

The 8.5 million km$^2$ Brazilian continental area harbors around 188 species of terrestrial, medium to large size mammals (*Abreu et al., 2021*), of which approximately 30% are considered threatened with extinction (*MMA, 2014*). We included a total of 190 mammalian taxa representing six orders: Carnivora (28 species), Cetartiodactyla (11 species; only terrestrial taxa), Cingulata (11 species), Perissodactyla (one species), Pilosa (12 species) and Primates (127 taxa). We included five small-sized mammals (*Cyclopes didactylus*, *C. ida*, *C. rufus*, *C. thomasi* and *C. xinguensis*) to cover all Xenarthran species in Brazil.

### Extinction risk assessment of Brazilian species

The assessment of the extinction risk of Brazilian fauna—conducted by the ICMBio—involves the scientific community (experts) and stakeholders worldwide. During this process, a specific form for each taxon is constructed, which contains information about its taxonomy, distribution, population, conservation, and threats and includes essential information for its extinction risk assessment. All the data is organized in the databank System for the Conservation Status Assessment of Brazilian Biodiversity—SALVE (*Sistema de Avaliação do Estado de Conservação da Biodiversidade*) (*ICMBio, 2022*). This Brazilian assessment process uses the IUCN Red List methodology for regional assessments, including criteria, categories, and quantitative and qualitative analyses (*ICMBio, 2013*; *IUCN, 2012*). For the analysis, we used the categories assigned to each species from the 2010–2014 Brazilian species extinction risk assessment and published in the Brazilian list of endangered species (*MMA, 2014*).

While the IUCN's global assessment uses nine categories—Extinct (EX), Extinct in the Wild (EW), Critically Endangered (CR), Endangered (EN), Vulnerable (VU), Near Threatened (NT) and Least Concern (LC)—for classifying species' risks of extinction (*IUCN, 2012*), at the regional level, such as the one used in the Brazilian assessment, two
additional categories are added: Regionally Extinct (RE) and Not Applicable (NA) (*IUCN, 2012*). Any relevant biological knowledge for the assessment is analyzed with rigorous quantitative criteria to meet one of these categories.

Population reduction, related to Criterion A, is one of the parameters used to identify whether a taxon has suffered/will suffer some level of population decline (number of mature individuals) (*IUCN, 2022*). However, most species lack direct observational or long-term populational studies, which would allow for a confident estimation of population reductions. Hence, other parameters, such as the decline in the Area of Occupancy (AOO), Extent of Occurrence (EOO), or habitat quality (*IUCN, 2022*), may be used to infer or suppose a population reduction.

As population reductions must be estimated within a time frame of three-generation lengths (3GL) or 10 years (whichever is longer) (*IUCN, 2022*), this parameter must be calculated according to the IUCN's definitions. According to the IUCN, Generation Length (GL) is defined as "the average age of parents of the current cohort (i.e., newborn individuals in the population)" (*IUCN, 2022*), reflecting the turnover rate of mature individuals in a population. The GL can be calculated in several ways. For this study, we adopted the following equation:

$$GL = AFR + (z * R_{span})$$

where: GL = Generation Length, AFR = Age of First Reproduction, $R_{span}$ = species reproductive life span, defined as the difference between the age at last reproduction and the age at first reproduction, and z is a constant, which "depends on survivorship and relative fecundity of young *vs* old individuals in the population" (*IUCN, 2022*). For primates, we adopted the GL estimated by the experts at the IUCN's Neotropical Primate Species Assessment Workshop in 2007. Since the essential information for calculating the GL of some species is absent, we adopted the GL estimated by *Pacifici et al. (2013)*, which is also recommended by the IUCN Red List Guidelines (2022).

## Habitat loss estimate

We used each taxon's geographical distribution, generated within the Brazilian species extinction risk assessment, to estimate habitat loss. This area consists of a polygon delimited by occurrence records (obtained through literature or personal communication with the assessors), and adjusted with the known biogeographical limits for the distribution of each taxon (*e.g.*, rivers or relief), according to the available literature or expert knowledge, when possible (*ICMBio, 2022*).

We used the MapBiomas Project—Collection 6 (2021), which has produced LULC data from 1985 to 2020, to estimate habitat loss. We first identified the LULC classes that constitute each species' suitable habitat; we then remapped the pixels within the distribution ranges to either 0 or 1 values, representing the non-habitat (0) or habitat (1) classes. Finally, after this pixel reclassification, we estimated habitat loss using the following equation:

$$HL = [1 - (HLY/HFY)] * 100$$

where HL = percentage of habitat loss, HFY = total habitat area of the first year of the time window (which is the sum of all pixels equal to 1 in the first year's map), HLY = total habitat area of the last year of the time window (sum of all pixels equal to 1 in the last years' map). We established 1985 as the first year of the time frame for the taxa with 3GL greater than 35 years as this is the oldest spatial data provided by the *MapBiomas Project (2021)*. Therefore, habitat loss might be underestimated for these taxa.

We developed the script for the habitat loss calculation using Google Earth Engine (GEE), and the code is available at: https://code.earthengine.google.com/?accept_repo=users/maributti/HabitatMammalsBR and https://github.com/MariellaButti/GEE-HabitatLossMapbiomas. This script allows the user to access the MapBiomas Project database online. Then, in order to calculate habitat loss, the user may select both the initial (ano_ini) and final (ano_fin) years—corresponding to the time window (3GL) for the target species, the scale (minimum = 30 m, which is the default resolution of MapBiomas Project), and the corresponding habitat(s) which is suitable for the target taxon (according to the MapBiomas Project Classes ID code). Additionally, it is possible to select upper and lower elevation limits to apply a mask to the habitat layer based on this range. Therefore, all habitat classes outside the selected range are reclassified as non-habitat (0). We used the NASA-SRTM (Shuttle Radar Topographic Mission) dataset (*Jarvis et al., 2008*) to implement this mask layer. In this study, we set up the limits out with Brazil's altitudinal range (−1,000, 4,000) as no species have elevation boundaries (Fig. 1).

We uploaded the taxon range file in vector format to perform the analyses on the GEE platform. The vector file must contain: .shp, .shx, .dbf and .prj extensions. Our habitat loss calculation script also makes it possible to draw a random polygon directly on to the map displayed on GEE in order to calculate the habitat loss in a specific area. Each taxon was analyzed separately, and outputs can be found in File S1. After running the script model, three data files containing the results are generated and are available for download: a CSV file, with all the requested information, and two raster files in .tiff, with the spatialized habitat of both the first and last years, corresponding to the taxon time window. It is also possible to save the files on Google Drive, in the cloud storage, or in the GEE Asset, as such, there is no need to save the files on the user's local computer.

## Data analyses

We performed descriptive statistics with the resulting habitat loss data. These analyses included mean, median, quartiles, standard deviation, and the maximum and minimum value for habitat loss according to Conservation Category (CR—Critically Endangered, EN—Endangered, VU—Vulnerable, NT—Near Threatened, LC—Least Concern, DD—Data Deficient, NA—Non-Applicable, and NE—Not evaluated); Conservation Status (Threatened—CR, EN, and VU; Non-threatened—NT and LC; None of the conservation status—DD, NA, and NE) and Taxa's Order (Carnivora, Cetartiodactyla/ Perissodactyla, Cingulata, Pilosa, and Primates). As the Perissodactyla order comprises only one species (*Tapirus terrestris*), we opted to include this taxon in the same group as the Cetartiodactyla order for the statistical analyses performed. To simplify the discussion, we divided habitat loss into three categories: High habitat loss ≥30% (considering the
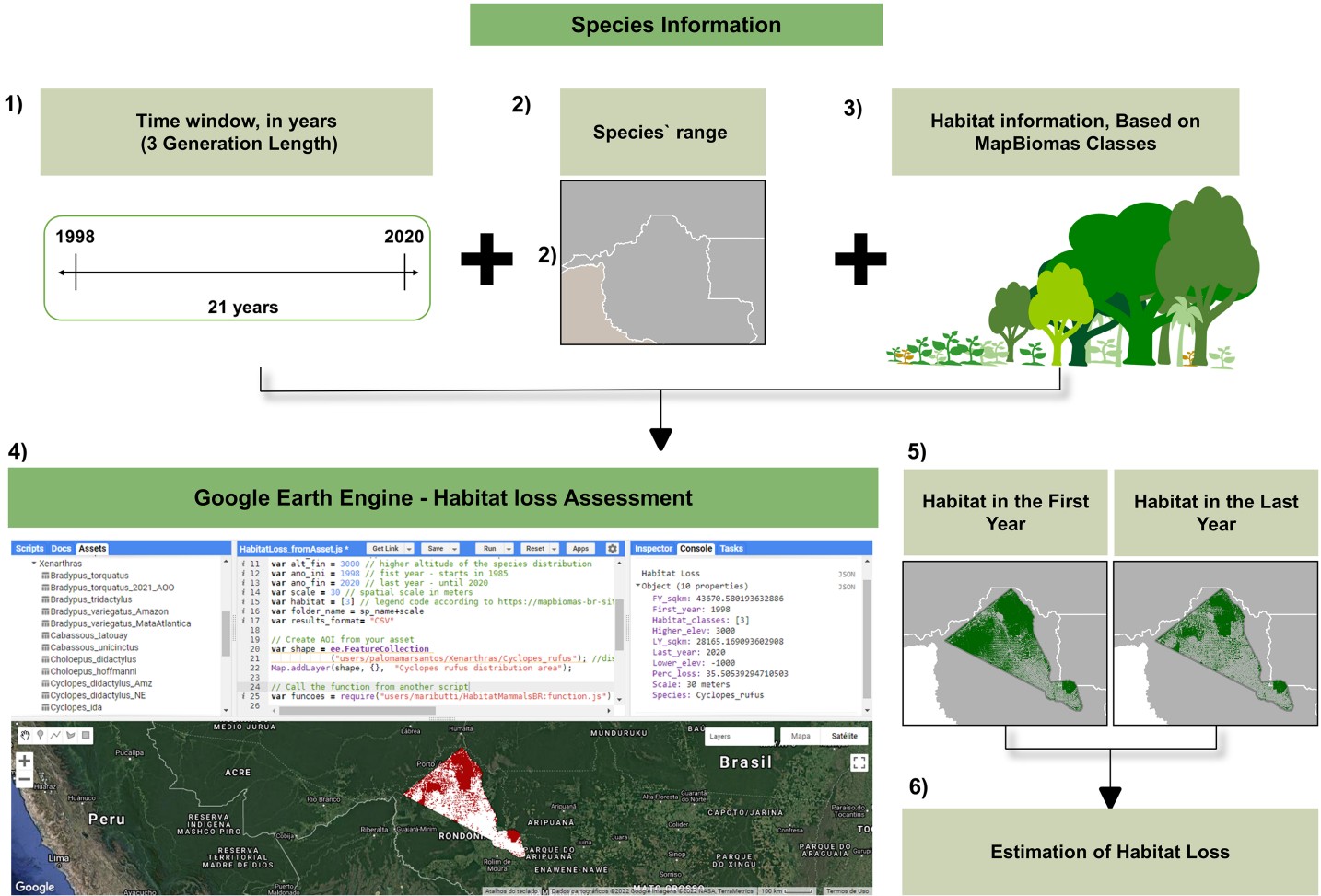

**Figure 1** **Workflow of inputs and outputs referent to the habitat loss estimation using our script through Google Earth Engine.** (1) Calculate the GL for the target taxon and the time window (=3GL). For this example, we used the GL of *Cyclopes rufus* = 7 years, then 3GL = 21 years. (2) Map the taxon range and upload the file to the GEE. (3) Identify the habitat classes suitable for the target taxon according to the MapBiomas Classes (4) Include all these input information on the script at GEE and run the model (5) Maps of both first and last years of the time window (3GL) are generated to estimate habitat loss (6) Habitat loss estimation output: the model run generates a CSV file with input settings and the results: first and last year habitat area (Km$^2$) and percent loss. Map data © 2022 Google Images © 2022 NASA, TerraMetrics.

minimum threshold of population reduction to classify the species in some of the threatened categories based on Criterion A); Moderate habitat loss ≥20% and <30% (considering the proximity to the previous threshold, which could lead the species to be categorized as Near Threatened); and Low habitat loss <20% (which could lead to a species categorization as Least Concern by Criterion A).

## RESULTS

We successfully developed an open script that estimates the habitat loss for each species of interest using a series of LULC maps from the MapBiomas Project through the GEE platform. We tested the application of the habitat loss calculation script on 190 mammalian taxa, covering different orders, habitat types, and conservation statuses.

The mean habitat loss was 6.46% (±8.23%), but with a high data variation, where the maximum and minimum values varied from 34% to −19.92% (Table 1). When observing the three target classes (Conservation Status, Conservation Category, and Order), the mean habitat loss followed the general trend of a high degree of variation, in terms of habitat loss (Table 1).

When comparing Conservation Status, the three categories presented a similar habitat loss (Fig. 2), where species with the status "None" suffered the greatest loss (Table 1). The NT category presented the highest habitat loss for the Conservation Category, followed by the "NA/NE" (Fig. 2; Table 1). Finally, habitat loss was similar for all five analyzed groups, with the order Carnivore presenting the highest values, followed by the Cingulata Order (Fig. 2; Table 1).

Most taxa included in this study (71%) suffered between 0–10% of habitat loss within their distribution, followed by 14% of the taxa that suffered between 10–20% of habitat loss. Only four taxa (2% of all included in this study) experienced habitat loss greater than 30%—three of which are already considered threatened with extinction and one which was previously categorized as Near Threatened (File S2). About 6.8% of the analyzed taxa's ranges (13 species) suffered between 20% and 30% habitat loss (Fig. 3), including five species (38.46%) that are already threatened with extinction (File S2). Less frequently, some taxa (6%–11 species) exhibited an increase in their habitat cover over 3GL (represented as negative values) (Fig.3; File S1).

## DISCUSSION

Our results revealed that many mammalian taxa suffered habitat losses to differing degrees and intensities over the past three generations, while only a few species experienced increases in habitat cover, highlighting the pervasive effects of habitat loss on species distribution and the potential effects on taxa persistence which have been previously presented in many studies (*Crooks et al., 2017*; *Heinrichs, Bender & Schumaker, 2016*). Overall, habitat loss occurred regardless of Conservation Category, suggesting that some of the current Non-threatened or Non-evaluated species might experience an uplisting (change to a higher threatened category) in the subsequent extinction risk assessment, solely based on these habitat loss analyses by inferring a consequent proportional population reduction. For example, *Tolypeutes matacus*, the southern three-banded armadillo, currently classified as NT, may suffer an uplisting due to the high habitat loss estimated for its range. This species only inhabits the Pantanal region, which has experienced severe wildfires over the past 2 years (2019–2020), compromising a significant number of refugees for this species (*Silva et al., 2020*).

Species that have been recently described but are still not evaluated, may also enter the list as threatened when first assessed. One example is the *Cyclopes rufus*, first described only 5 years ago (*Miranda et al., 2017*), which may be classified as Vulnerable since habitat loss within its range was estimated to be greater than 30% (File S2). This species lives in one of Brazil's most heavily deforested areas—the Arc of Deforestation in the Amazon, which is responsible for about 75% of the total habitat loss in the Amazonian region (*Fearnside, 2005*; *Oviedo, Lima & Augusto, 2019*). Similarly, considering the estimated

**Table 1 Descriptive statistics of the results from the habitat loss calculation script run on the 190 mammal taxa included in this study.**

|  |  | Mean | SD | Max | Min | Var |
|---|---|---|---|---|---|---|
| Conservation status/Category | Total | 6.48 | 8.22 | 34.07 | −19.92 | 53.99 |
|  | Threatened | 7.79 | 9.62 | 32.84 | −8.19 | 41.03 |
|  | CR | 11.10 | 14.75 | 32.84 | −1.12 | 33.96 |
|  | EN | 2.27 | 7.35 | 13.19 | −8.19 | 21.38 |
|  | VU | 9.54 | 8.79 | 31.88 | −1.17 | 33.05 |
|  | Non-Threatened | 5.22 | 7.06 | 31.77 | −19.92 | 53.99 |
|  | NT | 13.22 | 11.17 | 31.77 | −4.69 | 36.46 |
|  | LC | 4.24 | 5.75 | 24.44 | −19.92 | 44.36 |
|  | None | 8.06 | 8.53 | 34.07 | 0.40 | 31.37 |
|  | DD | 5.06 | 5.94 | 25.10 | 0.40 | 25.10 |
|  | NA/NE | 11.43 | 9.85 | 34.07 | 0.45 | 34.07 |
| Order | Carnivora | 8.81 | 6.53 | 26.17 | 0.95 | 25.21 |
|  | Cetartiodacyla/Perissiodactyla | 7.87 | 7.39 | 23.90 | 0.40 | 23.50 |
|  | Cingulata | 7.35 | 8.84 | 31.77 | 0.61 | 31.16 |
|  | Pilosa | 7.43 | 12.28 | 34.07 | −19.92 | 53.99 |
|  | Primates | 5.66 | 8.03 | 32.84 | −8.19 | 41.03 |

Note:
Mean, Mean value of habitat loss for that class; SD, Standard Deviation; Max, Maximum values of habitat loss for that class; Min, Minimum values of habitat loss for that class; Var, Variation between the minimum and maximum values.

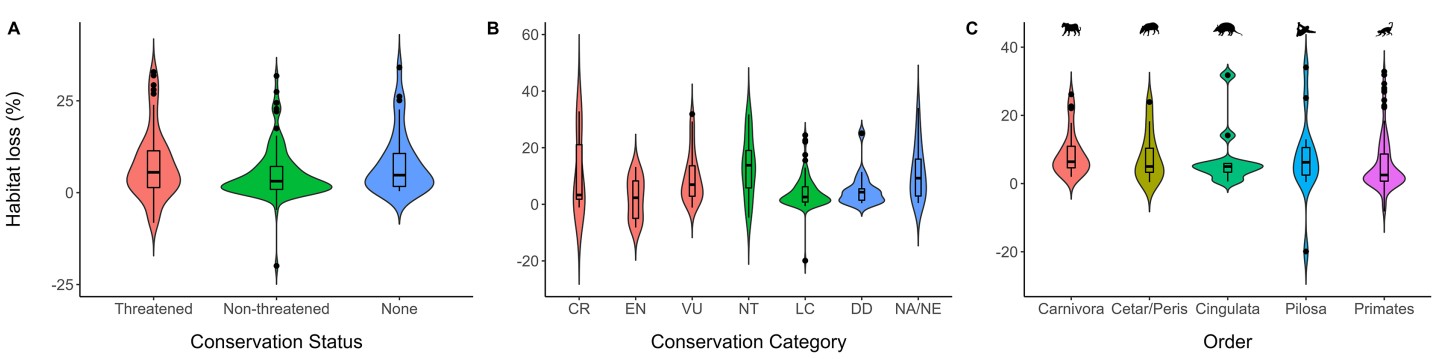

**Figure 2 Boxplots of the resulting habitat loss.** (A) Per conservation status categories (B) per conservation category (C) per order. The boxplot presents the minimum, maximum, the first and the third quartile, and the median. Data beyond the end represent the database outliers. Negative values refer to habitat gain. Cetar/Peris indicates the Cetartiodactyla and Perissodactyla orders, respectively.

habitat loss calculated in this study, the recently described *Leopardus munoai* and *Plecturocebus grovesi* may be classified as NT or threatened in the next extinction risk assessment (File S2).

Our results demonstrated that habitat loss also affects taxa that are already categorized as threatened. Nevertheless, despite being categorized as threatened, the vast majority of taxa only lost a moderate amount of their habitats. This may reflect taxa that experienced habitat loss too many years ago (*i.e.*, before the 3GL time window needed for the application of Criterion A) but have suffered from the secondary effects of habitat loss and

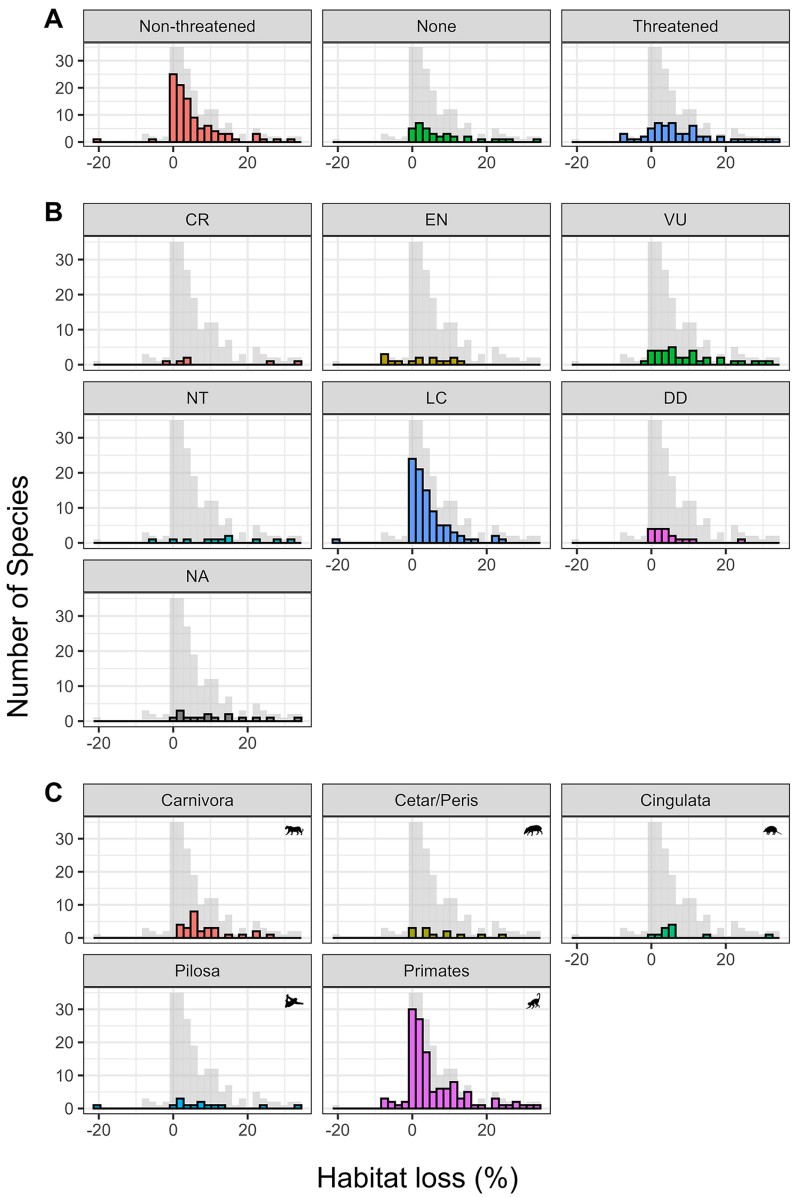

**Figure 3 Histograms showing the distribution of habitat loss for the three main classes analyzed.** (A) Per conservation status; (B) per conservation category; and (C) per order. The gray background histogram represents the distribution frequencies of habitat loss for all taxa. Cetar/Peris indicates the Cetartiodactyla and Perissodactyla orders, respectively.

may have been categorized as threatened under another criterion. Additionally, some taxa may inhabit the Atlantic Forest, which has already lost more than 70% of its forest coverage (*Rezende et al., 2018*; *Souza et al., 2020*), despite being protected by specific law regulations, which can make further deforestation difficult (Atlantic Forest Law, n° 11.428, December 22, 2006). Furthermore, some taxa may be categorized as threatened as a result of other factors, such as population reduction due to pathogens, competitors, human conflicts, pollutants, or parasites, or even due to small geographic ranges or population sizes, along with other threats like population decline. In such cases, habitat loss may only

be one of the threats leading to threatened categorizations. Indeed, for many taxa, habitat loss estimation must be accompanied by other types of data, such as population size reductions that are observed, estimated, or inferred by other sources—such as roadkill and poaching estimates, potential levels of exploitation, number of locations, among others (*IUCN, 2022*). Furthermore, most of these threatened taxa are included in National Action Plans, which may mitigate direct threats to the taxa.

The order Carnivora experienced the greatest habitat loss (Fig. 3C). This group is composed predominantly of predator species with low population densities and high spatial requirements. Therefore, this order is among the most threatened in Brazilian territory (*Mittermeier et al., 2013*). Following Carnivora, the orders Cetartiodicatyla and Perissodactyla, which compose the ungulates group, suffered the highest habitat loss intensity (Fig. 3C). These taxa include susceptible species with low reproduction, extended parental care, and high requirements for natural areas (*Beca et al., 2017*; *Jorge et al., 2021*; *Keuroghlian, Eaton & Longland, 2004*).

Contrarily to the Atlantic Forest biome, which has historically suffered extreme habitat loss, the Amazon and Cerrado biomes have suffered greater levels of natural habitat loss in the more recent past. The Amazon still concentrates a high proportion of natural remnants; however, it has lost more than 74 billion hectares of its forest habitat since 1985 (*MapBiomas Project, 2021*). The Cerrado, a Brazilian hotspot, has already lost 50% of its natural areas (*WWF- World Wide Fund For Nature, 2019*). The lack of specific legislation to protect this biome, which already exists for the Atlantic Forest, exacerbates the degradation of Amazon and Cerrado biomes, which is projected to continue in the future (*Soterroni et al., 2019*; *Strassburg et al., 2017*; *Velazco et al., 2019*). Conversely, all taxa that have experienced habitat expansion in their range inhabit the Atlantic Forest (File S2; Fig. S1). Over the past few years, this phytogeographic domain has experienced an increase in its natural areas (*Rezende et al., 2015*; *Júnior, Santos & Eugenio, 2012*), due, in part, to specific environmental laws, social-economic shifts, and conservation pacts (*Baptista & Rudel, 2006*; *Fundação SOS Mata Atlântica, 2021*).

## Implications for conservation and management

The data generated from our habitat loss calculation script will undoubtedly support subsequent assessments of terrestrial mammalian taxa in Brazil. Aside from supporting extinction risk assessments, our open data framework results may present other conservation implications. The generated raster files may be post-processed to identify the geometric patterns of habitat loss (Figs. S2 and S3). When analyzing deforestation data from *PRODES (2016)*, *Maurano, Escada & Renno (2019)* identified deforestation patterns with distinct levels of complexity, which can be observed in this study for Amazonian species: diffuse, linear, regular geometric, multidirectional, herringbone and consolidated, of which the latter three are more complex than the former. These different patterns are related to distinct patterns of anthropic occupation and, will consequently, have different effects on species (*Maurano, Escada & Renno, 2019*). Similarly, with the raster files generated by this method, it is possible to quickly understand the importance of Protected Areas and Indigenous territories to safeguard biodiversity (Fig. S4).

Additionally, the information generated can be a robust and essential aid for elaborating and implementing Conservation Action Plans. The ICMBio also coordinates the National Actions Plans for Conservation, which focus on conservation strategies for threatened species and contemplate the near-threatened ones. These plans are public policies that are constructed and agreed upon by diverse stakeholders, which aim to identify and guide priority actions to reverse threats to wildlife populations and environments to ensure their survival (Normative Instruction 21/2018; *Ministry of the Environment of Brazil/Chico Mendes Institute for Biodiversity Conservation, 2018*). Therefore, our results may assist in identifying critical and priority areas for protection and restoration, thus directing efforts to avoid increases in habitat loss or to minimize the effects of habitat loss of persistent threatened species through actions by law enforcement, protected areas, habitat connectivity, and enrichment, population management, conservation, education, among others.

The IUCN Red List website (*IUCN, 2020*) provides guidelines, tools, and spatial data that assessors may apply in order to map species' ranges. Many of these tools are developed into proprietary software and run locally on the user's machine. Thus, using our script on the GEE platform as a new tool for estimating habitat loss could facilitate spatial analysis in Brazilian national assessment workshops and even for other South American countries or global assessments. We anticipate that this open data framework has the potential to further qualify the evaluation process in different regions as it is fast processing, accessible free of charge for academic and research purposes and does not require local computer memory or higher processing requirements (*Arruda et al., 2021*; *Gorelick et al., 2017*; *Souza et al., 2020*).

Specialists have improved the extinction risk assessment of biodiversity. However, many taxa still lack essential biological or long-term studies to directly subsidize population reduction estimations. Therefore, assessments are often based on poorly scientifically-founded "best guesses". Using new spatial cloud-based technologies can help improve assessments and, ultimately, support decision-making for biodiversity conservation. This approach does not replace direct observations on the impacts on wildlife populations (*e.g.*, *Tomas et al., 2021*). However, it offers complementary tools that are often more widely and quickly applicable, to help understand how much and where habitat loss affects different species and distinct groups. Furthermore, this is especially important for improving conservation assessment processes and establish and implement more efficient conservation measures, particularly when rapid responses are required, and limited resources are available.

## CONCLUSIONS

In this study, we successfully used the accessible dataset of the MapBiomas project through the Google Earth Engine platform to generate a user-friendly, open-source, and replicable script to calculate habitat loss for terrestrial mammalian taxa, generating accurate and reliable data to support extinction risk assessments. The script's usefulness was demonstrated by estimating habitat loss for 190 mammalian species in Brazil. Improving habitat loss analyses is a crucial step for improving the conservation status assessment for

Brazilian mammals and other terrestrial taxa in the future. Additionally, our results provide new information which can guide future research and conservation efforts. Finally, the script and protocol we present here can be directly applied or adapted for use in other similar assessments while remaining open to continuous development.

## ACKNOWLEDGEMENTS

We are extremely grateful to the MapBiomas Project for the MapBiomas Award 2022 on the application of its database on public policies offered to a preliminary version of this article. We also would like to acknowledge the network of collaborators who helped in compiling and validating the occurrence records and defining the distribution polygons of all terrestrial mammals of Brazil during the national conservation status assessment, especially the Coordinators of Taxa who were crucial for the success of this process. Mariella Butti would like to especially thank Raquel Costa da Silva, Renan Lieto A. Ribeiro, Marina Portugal, Beatriz Garcia, Maria Elisa de F. Morandi for reviewing the Carnivore and Ungulates distributions; and Paula Condé, Livia Almeida, Ronaldo Morato, Rogerio Cunha and Marcelo Magioli for supporting and encouraging her into this issue.

### Funding

The Instituto Chico Mendes de Conservação da Biodiversidade (ICMBio) supported the personnel and infrastructure to develop this project. Paloma M Santos received a research scholarship from ICMBio and the National Council for Scientific and Technological Development—CNPq (Grant Numbers 350057/2020-6) and from Programa de Capacitação Institucional–PCI/INMA (Grant Numbers 317795/2021-0; 300893/2022-1) and Gabriela Ludwig from ICMBio, Fundação de Apoio à Pesquisa—FUNAPE (Grant Number 6774) and CNPq (350404/2018-6). The funders had no role in study design, data collection and analysis, decision to publish, or preparation of the manuscript.

### Grant Disclosures

The following grant information was disclosed by the authors:
The Instituto Chico Mendes de Conservação da Biodiversidade (ICMBio).
ICMBio and the National Council for Scientific and Technological Development—CNPq: 350057/2020-6.
Programa de Capacitação Institucional–PCI/INMA: 317795/2021-0; 300893/2022-1.
Gabriela Ludwig from ICMBio, Fundação de Apoio à Pesquisa—FUNAPE: 6774.
CNPq: 350404/2018-6.

### Competing Interests

The authors declare that they have no competing interests.

## Author Contributions

- Mariella Butti conceived and designed the experiments, performed the experiments, analyzed the data, prepared figures and/or tables, authored or reviewed drafts of the article, and approved the final draft.
- Luciana Pacca conceived and designed the experiments, performed the experiments, analyzed the data, prepared figures and/or tables, authored or reviewed drafts of the article, and approved the final draft.
- Paloma Santos conceived and designed the experiments, performed the experiments, analyzed the data, prepared figures and/or tables, authored or reviewed drafts of the article, and approved the final draft.
- André C. Alonso conceived and designed the experiments, performed the experiments, analyzed the data, prepared figures and/or tables, authored or reviewed drafts of the article, and approved the final draft.
- Gerson Buss conceived and designed the experiments, performed the experiments, analyzed the data, prepared figures and/or tables, authored or reviewed drafts of the article, and approved the final draft.
- Gabriela Ludwig conceived and designed the experiments, performed the experiments, analyzed the data, prepared figures and/or tables, authored or reviewed drafts of the article, and approved the final draft.
- Leandro Jerusalinsky conceived and designed the experiments, performed the experiments, analyzed the data, prepared figures and/or tables, authored or reviewed drafts of the article, and approved the final draft.
- Amely B Martins conceived and designed the experiments, performed the experiments, analyzed the data, prepared figures and/or tables, authored or reviewed drafts of the article, and approved the final draft.

## Data Availability

The scripts, example maps, the description of habitat loss output table fields, and the input data used to run the analysis and results for each species are available in the Supplemental Files.

## Supplemental Information

Supplemental information for this article can be found online at http://dx.doi.org/10.7717/peerj.14289#supplemental-information.

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
