# Peer review of "Habitat loss estimation for assessing terrestrial mammalian species extinction risk: an open data framework"

_PeerJ, doi:10.7717/peerj.14289_

## Round 0.1 · original submission · Minor Revisions

Both reviewers agree that your article can be published after minor revision. They provided an excellent and comprehensive set of suggestions that you should use to improve the manuscript. Give special attention to citations (always cite the original source of the statement you want to support) and language. In fact, several sentences are a bit difficult to follow, and both reviewers suggested you ask a proficient speaker of English to review your paper carefully for both style and grammar. I agree with them.

Please ensure that all review and editorial comments are addressed in a response letter. Any edits or clarifications mentioned in the letter are also inserted into the revised manuscript where appropriate.

I look forward to seeing an improved version.

Reviewer 1 ·

Basic reporting

This paper provides a convenient technique for rapidly automating habitat loss calculations, with the goal of improving threat assessments, particularly of mammals in Brazil. The goals, methods, and conclusions are well aligned, and the purpose and outcome of the paper is clear. The research is good, and the explanation of the data/code appears user-friendly (if the user is familiar with GEE). My main concern is really with the grammar and writing. One frequent issue is use of general words like “this” and “it” rather than explicitly describing the item or process being referred to, which makes some of the background and methodology sections challenging to follow. There are a few specific, but non-exhaustive examples that I note below throughout the text that could use clarification or re-phrasing.

I would recommend having the manuscript reviewed by a fluent English-speaking colleague, to ensure that readers are fully able to follow the explanation of the work before recommending it for publication. As mentioned, I have listed a few of the grammar and writing issues that I noted, as well as a couple of additional minor questions and clarifications about the methods.

Abstract/Intro:
Line 27: the use of “this” at the beginning with no context is vague.
Line 32: what species distribution and ecology? I think it’s useful to be more specific here
Line 77: considers rather than comprehends may be more accurate
Line 87-88: this sentence reads oddly

Methods: Section 2.3

It is unclear if the authors generated the polygon or if they retrieved the polygon from a different source and are describing that source’s methods.

Line 216: The description of the Supplemental File is confusing; I would recommend to either just say “Each taxon was analyzed separately, and outputs can be found in Supplemental File 1” or describe in slightly more detail what is meant by “ten output parameter information”.

Methods: Section 2.4
Line 233: I appreciate the honesty of saying that the categories are arbitrary, but the authors then describe a reason for using the categories, so it seems there is justification. It may be better to rework this to explain the categories.
Results:
Line 244: Assuming this is percent, it should probably be stated as such.

Discussion:
Line 272: Not sure what is meant here by upgrade in category. Being listed as more threatened? You later refer specifically to uplisting, so I would recommend using consistent terminology if you do mean the same thing.
Line 293-4: This reads like an incomplete sentence and I am not entirely sure what its goal is.
Line 312: Unsure about the use of the word possibly here; is this not a result that was generated from this study?
Line 315: Authors use natural habitat/natural areas throughout the paper, but here use territory which has slightly different connotations about people. If this is meant to refer to the habitat loss, consistent terminology would be better.
Line 363: complementary, not complimentary

Conclusions:
The first sentence could be improved to better summarize the purpose of the article. I’m unsure what is meant by qualified in line 370.
Figures/Tables:
There is a typo in Figure 1, “Habitat information, Based on MapBiomas Classes*”

Experimental design

Methods Section 2.3:

Are there limitations on the ano_ini and ano_fin variables or a way of ensuring that the 10 year or 3GL minimum is being enforced? If this is to be used to contribute to assessments, it could be a useful parameter to ensure.

Validity of the findings

In the raw data file (Supplemental File 2), the conservation status tag assigned to DD is “Non-threatened” while in the paper, it states that DD species are grouped with NE/NA, which is tagged as “Not evaluated” (or maybe “None”, this is also slightly unclear/inconsistent). Is this a new assessment of the DD species after having done the habitat loss evaluation, or is there a reason these are being assumed “non-threatened”? This should be clarified either in text or in the data, and if DD is incorrectly tagged, the various descriptive and summary statistics of Table 1, Figure 2, and Figure 3 should be reconfirmed to make sure that the information on “None” vs “Non-threatened” is accurate.

Reviewer 2 ·

Basic reporting

The manuscript is self-contained and presents its key questions and results in a clear manner. It is organized in clear sections with a suitable structure and the figures and tables are well prepared and presented. The analytical scripts used for the work are openly shared and the data used for the work, namely that from MapBiomas, are openly available. The text is well supported by a prolific use of references to previous work but in sections I find the references cited could be more relevant. For example, the citation to the latest IPCC report as a means to support the statement that only 16% of natural habitats remain (lines 47-48) is in my opinion inadequate, and could be replaced to a reference to the 2019 IPBES report instead. I'd recommend the authors to consider more care when citing previous work to support a statement by ensuring the cited reference clearly supports it. Finally, the language cintains some typos or grammatical errors that could be improved to ensure that an international audience can clearly understand your text. Some examples where the language could be improved include lines 96-99, 108-110, 230-232, 293-295, as the current phrasing is not correct or makes comprehension difficult.

Experimental design

The research presented in this work is clearly within the Aims and Scope of the journal. The question is well defined and focuses on how much habitat has been lost for different mammal species in Brazil over the last 35 years. This question is relevant given the importance of this assessment to assist estimates of species extinction risks and helps to fill an information gap related to the potential reduction of the populations of the assessed species through habitat destruction. The work was seemingly developed to a high technical standard and the methods are reported in thorough detail, including the open sharing of the computational scripts used to carry out the analysis, which allows future replication of the work.

Validity of the findings

The findings are straightforward to interpret, robust and statistically sound, highlighting how most of the assessed species lost some area of their habitat during the period assessed. Nevertheless, the authors also present evidence that a few species gained area of habitat, making the assessment honest and comprehensive. The authors discuss the results of their work in adequate detail and the conclusions derived from the results are well supported.

Additional comments

I provide a few additional minor comments on specific sections of the manuscript may be useful for the authors to consider further:

Lines 27-28: This is a very strong claim, particularly when so many biological groups. I'd suggest replacing "the most severe" with "a severe" which retains the emphasis but avoids exaggeration.

Lines 33-34: I find this sentence unclear with regards to what this means and why it was chosen. I'd suggest replacing it with: We considered the period of three generations length to assess habitat loss in line with Red List assessment criteria.

Lines 66-67: The Aichi Targets have ceased in 2020, and thus this reference to them is perhaps a bit outdated. I'd suggest referring to the draft of the post-2020 biodiversity targets, which is likely to be more relevant going forward. You can find the latest draft of the proposed goals here: https://www.cbd.int/doc/c/abb5/591f/2e46096d3f0330b08ce87a45/wg2020-03-03-en.pdf

Lines 96-99: This sentence doesn't read well, it seems as if the Brazilian territory lost area, not natural habitats. Consider rephrasing; I'd suggest the following: "An area of approximately 82Mha of natural habitats was lost in Brazil between 1985 and 2020, including a notable increase in deforestation in recent years in almost all phytogeographic domains, directly affecting multiple taxa."

Lines 140-141: Used for what? Do you mean you consider the latest assessment to identify the species that are the focus of the study? Please clarify.

Line 152: Data and information are redundant here, please consider removing one.

Line 156-158: It would perhaps be relevant here to highlight what the additional categories are and their relevance in the context of this work. This is particularly important because the reference you cite is in Portuguese language and thus potentially not readily interpreted by the reader(s).

Line 179: Red List is usually referred to with capital letters, and the authors adopt this style across most of the manuscript but not here.. Please consider revising this and being consistent throughout the manuscript.

Line 316: Specific legislation for what? Can you please clarify?

---

## Round 0.2 · accepted · Accept

Thanks for your resubmission. I am glad to accept the updated version of your manuscript for publication.